# Salinity as a Key Factor in Structuring Macrophyte Assemblages in Transitional Water Bodies: The Case of the Apulian Coastal Lagoons (Southern Italy)

Andrea Tursi [1,2], Anna Lisco [1], Giovanni Chimienti [2,3], Francesco Mastrototaro [1,2], Nicola Ungaro [4] and Antonella Bottalico [1,*]

1 Department of Biosciences, Biotechnologies and Environment (DBBA), University of Bari "Aldo Moro", Bari 70125, Italy; andrea.tursi@uniba.it (A.T.); annalisco87@gmail.com (A.L.); francesco.mastrototaro@uniba.it (F.M.)
2 CoNISMa LRU, National Interuniversity Consortium for Marine Sciences, Rome 00196, Italy; giovanni.chimienti@uniba.it
3 Red Sea Research Center, Biological and Environmental Science and Engineering Division, King Abdullah University of Science and Technology (KAUST), Thuwal 23955-699, Saudi Arabia
4 Apulian Regional Agency for the Environmental Prevention and Protection, Bari 70126, Italy; n.ungaro@arpa.puglia.it
* Correspondence: antonella.bottalico@uniba.it; Tel.: +39-0805442163

**Abstract:** Coastal lagoons are dynamic transitional water ecosystems hosting valuable biological communities, including rich and diverse macrophyte assemblages. Aquatic macrophytes must cope with large fluctuations of environmental conditions on a spatial and seasonal scale. Salinity is one of the most variable parameters, changing from nearly freshwater to hypersalinity, and it is known to have a strong influence on the composition and structure of macrophyte assemblages. This study is focused on the effect of salinity on macrophyte communities of the eight most important coastal lagoons of Apulia (south-eastern Mediterranean Sea). A set of eleven transitional water body types (*sensu* Water Framework Directive) were allocated in a range of meso- to hyperhaline lagoons. Macrophyte sampling was carried out between 2011 and 2019, and a total of 324 samples (18 sampling stations × 2 seasons × 9 years) was analyzed. Then, macrophyte occurrence in each transitional water body (T-WB) was expressed as frequency values (%) and assemblages were compared to assess any similarity in relation to four salinity classes (mesohaline, polyhaline, euhaline and hyperhaline). Species richness varied according to the salinity class, being much higher in polyhaline and euhaline T-WBs and strongly decreasing at the extremes of the salinity range (mesohaline and hyperhaline T-WBs). Moreover, the statistical analysis showed a high resemblance of macrophyte assemblages of T-WBs within the same salinity class, which shared a great number of species. Four distinct macrophyte communities were distinguished, reflecting the salinity conditions of different T-WB types and confirming the effectiveness of a lagoon typology based on this descriptor.

**Keywords:** transitional water ecosystems; coastal lagoons; biodiversity; typology; aquatic vegetation; abiotic factors; benthic communities; conservation; environmental variability

## 1. Introduction

Transitional waters, including coastal lagoons, saline lakes, river estuaries and deltas, are dynamic and heterogeneous ecosystems placed at the interface between water and terrestrial environments receiving variable amounts of freshwater [1,2]. Due to their geomorphological and hydrological features, such as the shallow depth, partial confinement, limited extension, variation of freshwater inputs and the exchange flows with marine waters, these environments frequently undergo strong fluctuations in physical and chemical parameters such as salinity, temperature, turbidity, dissolved oxygen and pH [3]. Moreover, transitional waters are characterized by high trophic inputs and rapid biogeochemical

cycles, being highly sensitive to eutrophication events [4–6]. Such conditions select peculiar macrophyte communities well-adapted to the variability of abiotic factors [7]. Besides opportunistic species, these ecosystems are naturally colonized by aquatic angiosperms and macroalgae with remarkable ecological value [8,9]. Submerged macrophytes represent important primary producers and many species are considered ecosystem engineers, providing several ecosystem goods and services [10], including substrate stabilization, erosive processes reduction [11], acidification and eutrophication mitigation, as well as waters and atmosphere oxygenation [12]. Healthy macrophyte associations host rich and diverse communities of aquatic invertebrates, fish and birds, which rely on these habitats as spawning, nursery, feeding, and refuge areas [13,14]. Among transitional waters, coastal lagoons are listed as priority habitats in Annex I of the Habitat Directive (92/43/EEC) [15] and are protected by several national and international agreements (Natura 2000 network; Ramsar convention). Hence, conservation and sustainable management of such habitats are strongly advocated [16]. Moreover, macrophytes can be used as ecological indicators to assess the quality of water bodies due to their ability to respond to environmental changes [17–20]. As a consequence, they are among the biological elements proposed in the European Water Framework Directive (WFD; 2000/60/CE) [21] as quality elements for the classification of the ecological status of transitional water bodies (T-WBs). In Italy, the use of the Macrophyte Quality Index (MaQI) [22–24] is currently compulsory under the national law (Italian Ministerial Decree 260/2010).

Although the crucial role of macrophytes in transitional waters is well-known, there is still scarce knowledge about the species' ecological traits and their responses to the main abiotic factors. Salinity is known to be one of the most variable parameters in T-WBs, and it can be a key factor in modelling the biodiversity of macrophyte assemblages [7,25–27]. Some T-WBs are nearly fresh, with a mean salinity below 1 PSU [28], while hypersaline transitional waters can reach mean salinity values of 75 PSU [29,30]. Moreover, within the same T-WB, salinity can also show a great range of variability during the year, even reaching a difference of 100 PSU between winter and summer [31]. The presence of a strong genetic adaption of species to salinity regimes of transitional waters has been demonstrated by previous studies, proving the crucial role of this abiotic factor in determining the macrophyte distribution [32–34].The variation pattern of biodiversity along the salinity gradient was already discussed in the past by the Remane paradigm [35], which stated a continuous decline of benthic species richness with the diminishing of salinity (minimum values at about 6 PSU). This trend was also confirmed by Adams et al. (1992) [36], which observed the highest macrophyte diversity between 25 and 30 PSU, while it declined rapidly above 35 PSU, and was recently confirmed by Schubert et al. (2011) [37], according to which the general macrophyte biodiversity declines from euryhaline to oligohaline waters, as well as in hyperhaline conditions. However, there is still a lack of knowledge about the presence of possible analogies in the macrophyte species composition of T-WBs with similar salinity ranges.

In the Mediterranean ecoregion, four types of transitional waters (estuaries, deltas, microtidal and non-tidal coastal lagoons) are present and, according to WFD, a typological scheme was defined by each country. Coastal lagoons were distinguished on the base of their shape (as surface area), tidal range and salinity, as indicated in System B of WFD, in the Italian lagoon typology [38]. Salinity thresholds were derived from the Venice System [39]. In particular, the Italian lagoon types were classified on two levels of tidal range (> or <50 cm), two of surface area (>2.5 km$^2$ or between 0.5 and 2.5 km$^2$) and five of water salinity (oligohaline: <5 PSU, mesohaline: 5–20 PSU, polyhaline: 20–30 PSU, euhaline: 30–40 PSU and hyperhaline: >40 PSU) [38,40], according to the indications given by the European Common Implementation Strategy (CIS) working groups (CIS Guidance n. 5).

In the Mediterranean context, more than 130 transitional water systems are present in Italy, with a surface area greater than 1700 km$^2$ [41]. The largest coastal lagoons are located along the northern Adriatic coasts, but the highest number is concentrated in the southern part of the Italian peninsula. Our work is focused on the salinity influence on

the macrophyte species composition of the eight most important and wider transitional basins of the Apulian region (south of Italy) [42]. The study falls within the monitoring and assessment of the ecological status of T-WBs by MaQI, as required by Italian legislation (Italian Ministerial Decree 260/2010). All the investigated T-WBs are located in non-tidal coastal lagoons of small and medium size, and are distributed into four salinity classes, from mesohaline to hyperhaline [43]. We assessed the presence of any relation among macrophyte assemblages in T-WBs of the same salinity class, as well as their species richness, aiming at proving the role of salinity in structuring macrophyte communities. The validity of the Apulian lagoon typology, mainly based on the salinity descriptor, was also checked as a case study in the Mediterranean ecoregion.

## 2. Materials and Methods

The study was based on the identification and classification of the coastal lagoons of Apulia following the current Italian legislation (Italian Ministerial Decree 131/08). According to it, the investigated area included eight lagoons, two for each salinity class: Cesine (CE) and Torre Guaceto (TG) classified as mesohaline; Lesina (LE) and Varano (VA) as polyhaline; Porto Cesareo (PC) and Mar Piccolo (MP) as euhaline; Margherita di Savoia, only for the part known as "Vasche Evaporanti—Lago Salpi" (LS); and Punta della Contessa (PU) as hyperhaline (Figure 1). A total of 11 T-WBs were considered. In almost all cases, each lagoon comprised a single T-WB, with the exception of LE, divided into three different T-WBs (LE1, LE2 and LE3) and MP divided into two T-WBs corresponding to two well-distinct inlets, called the First and Second Inlet (MP1 and MP2, respectively).

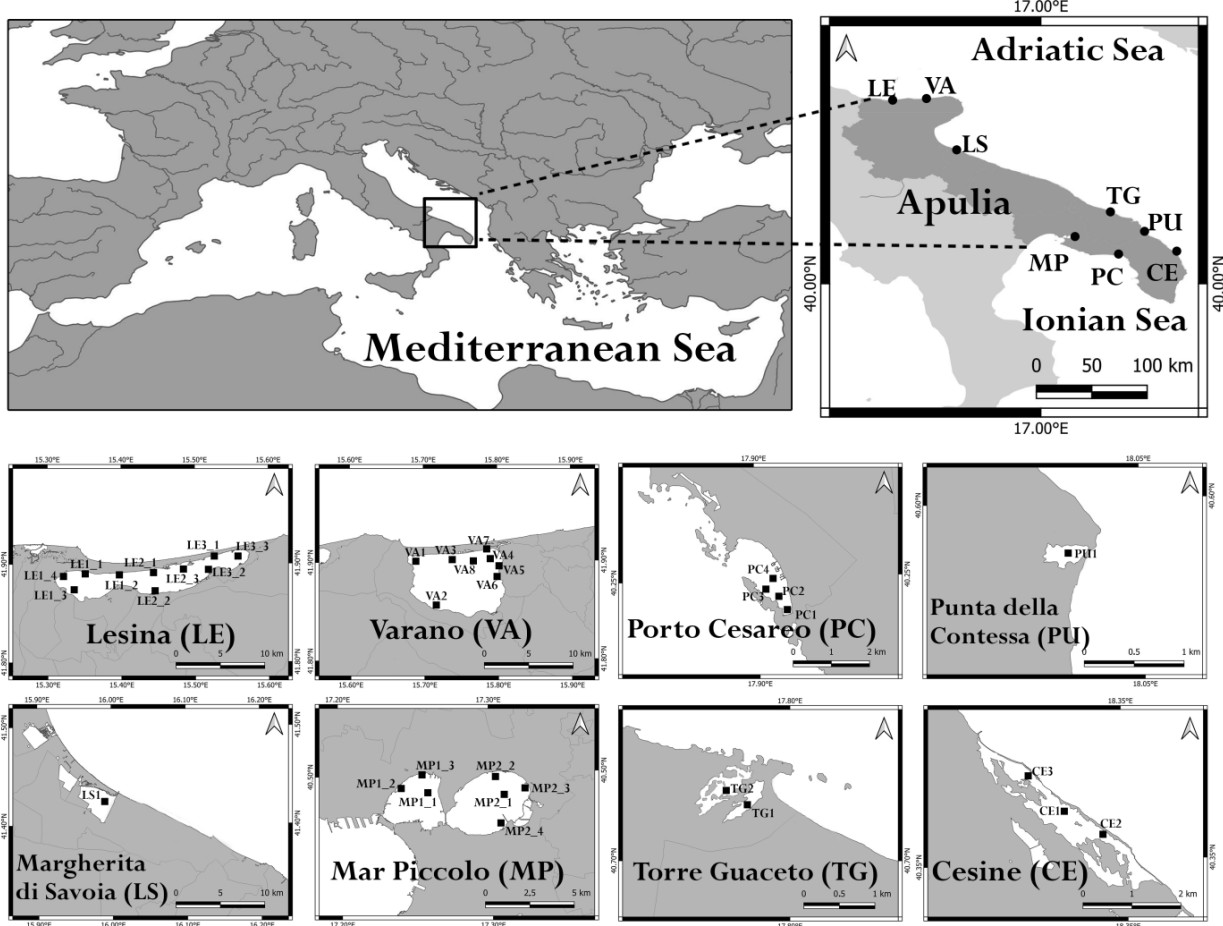

**Figure 1.** Map of the study area showing its position in the Mediterranean Sea, the location of the transitional water basins and the sampling stations placed in each transitional water body (*sensu* WFD).

Macrophyte sampling was carried out from 2011 to 2019, during spring and autumn, for a total of 18 seasonal surveys. Sampling stations were distributed taking into account the morphology, spatial heterogeneity and extension of each T-WB (Figure 1; Table 1). Only one station was chosen as representative of the hyperhaline basins (PU and LS), which are choked lagoons with homogeneous ecological conditions. Two stations were sampled in TG, a small basin almost entirely dominated by *Chara baltica* (Hartman) Bruzelius. Three stations were placed in CE, inside the main perennial pond called the Pantano Grande (0.68 km$^2$). Four stations were selected to cover PC, a bay in open connection with the sea; meanwhile, to characterize the larger basins of LE, VA and MP, a higher number of stations was required (ten, eight and seven, respectively). In detail, for the different T-WBs of LE and MP, it ranged between three (in LE2, LE3, MP1) and four stations (in LE1 and MP2). The southern part of VA was not investigated due to the lack of vegetation (Bottalico, pers. obs.) In total, 324 samples (18 stations × 2 seasons × 9 years) were analyzed. Samples were collected according to the monitoring protocols by the Italian Institute for Environmental Protection and Research [44] and as required by MaQI application [22,23]. Macrophytes were preserved in 4% buffered formalin/seawater and determined at specific and intra-specific levels in the laboratory by means of a Leica MZ 7.5 stereo-microscope (Leica, Wetzlar, Germany) and a light microscope Olympus BX-40 (Olympus, Melville, NY, USA). For morphological observations, sections of the thalli were obtained by free-hand cutting or with a DSK-1000 vibratome (Dosaka, Kyoto, Japan). Some samples of uncertain identification were also preserved in silica gel for molecular analyses. The up-to-date nomenclature of the identified taxa followed the AlgaeBase [45].

**Table 1.** Coordinates, extension (km$^2$), depth range (m) and salinity class (PSU) of each T-WB.

| WB | Coordinates | Surface Area (km$^2$) | Depth (m) | Salinity Class (PSU) * |
|---|---|---|---|---|
| Torre Guaceto (TG) | 40°42′51.34″ N–17°47′42.91″ E | 1.2 | 0.4–0.6 | Mesohaline (5–20) |
| Cesine (CE) | 40°21′33.46″ N–18°20′09.30″ E | 0.7 | 0.2–0.8 | Mesohaline (5–20) |
| Lesina (LE1) | 41°53′12.64″ N–15°21′15.65″ E | 18 | 0.7–2 | Polyhaline (20–30) |
| Lesina (LE2) | 41°53′01.23″ N–15°27′20.15″ E | 17 | 0.7–2 | Polyhaline (20–30) |
| Lesina (LE3) | 41°53′57.20″ N–15°31′00.45″ E | 16 | 0.7–2 | Polyhaline (20–30) |
| Varano (VA) | 41°52′43.65″ N–15°44′35.42″ E | 60.5 | 0.5–5 | Polyhaline (20–30) |
| Porto Cesareo (PC) | 40°14′31.80″ N–17°54′32.82″ E | 2 | 0.3–5 | Euhaline (30–40) |
| Mar Piccolo (MP1) | 40°29′19.68″ N–17°15′29.51″ E | 9.7 | 0.5–12 | Euhaline (30–40) |
| Mar Piccolo (MP2) | 40°29′22.92″ N–17°18′29.18″ E | 11 | 0.5–12 | Euhaline (30–40) |
| Margherita di Savoia (LS) | 41°25′27.34″ N–15°59′53.29″ E | 8.5 | 0.4–0.6 | Hyperhaline (>40) |
| Punta della Contessa (PU) | 40°35′42.31″ N–18°02′30.05″ E | 2 | 0.4–1 | Hyperhaline (>40) |

\* for the first identification of T-WBs according to the Italian Ministerial Decree 131/08.

The macrophyte occurrence in each water body was expressed as frequency values (%), representing the ratio between the number of samplings where the species were found and the total number of samplings carried out in a given T-WB. Then, macrophyte assemblages were compared to assess any relation between the species presence and abundance with salinity classes. To test the salinity influence on macrophyte communities, Cluster Analysis and *non*–Metric Multidimensional Scaling (*n*–MDS) were performed using the software Past 4.03. For the Cluster Analysis, the Paired Group Algorithm (UPGMA) and the Bray–Curtis dissimilarity matrix were used, as they are particularly reliable for comparisons in species composition. The *n*-MDS was also performed using the Bray–Curtis index. The non-parametric ANOSIM (Analysis of Similarities) was tested to verify the data significance (*p*-values and *R*-values). Then, the SIMPER test (Similarity Percentages) was run using the software Primer 5 (version 5.2.9) to evaluate the contribution of each species (%) to

the similarity among T-WB groups identified by clustering (cut-off at 90%). The Species Richness index (S), representing the total number of species counted in each T-WB, was also calculated. The normality of data distribution was assessed with the Shapiro–Wilk test using Past 4.03, then the significance of the differences in Species Richness among the four different salinity classes was assessed by performing the ANOVA (Analysis of Variances) parametric test.

## 3. Results

A total of 171 species belonging to Magnioliophyta (3.5%), Charophyta (3.5%), Ochrophyta-Phaeophyceae (8%), Chlorophyta (33%) and Rhodophyta (52%) were identified (Table 2). A considerable part of the species was only sporadically found in one or few T-WBs, with very low frequency. On the other hand, several species of Chlorophyta, mainly Ulvales and Cladophorales, stably colonized different T-WBs. Some taxa also resulted to be very abundant in certain circumstances, acting as structuring species. This is the case of some species of phanerogams, such as *Cymodocea nodosa* (Ucria) Ascherson, *Ruppia spiralis* Linnaeus *ex* Dumortier and *Zostera noltei* Hornemann, as well as perennial brown algae, such as *Gongolaria barbata* (Stackhouse) Kuntze, which colonized wide areas in the presence of salinity values suiting with their ecological traits.

**Table 2.** Taxonomic list of the species found during the monitoring periods (2011–2019), grouped in orders inside each phylum. The table reports the frequency values abundance (%) of the taxa in the four different types of T-WBs according to salinity classes (HH: hyperhaline; EH: euhaline; PH: polyhaline; MH: mesohaline). +: 1–24%; ++: 25–49%; +++: 50–74%; ++++: 75–100%.

| | HH | EH | PH | MH |
|---|---|---|---|---|
| **MAGNOLIOPHYTA** | | | | |
| **Alismatales** | | | | |
| *Cymodocea nodosa* | | +++ | + | |
| *Ruppia spiralis* | ++++ | | ++ | ++ |
| *Zannichellia palustris* | | | | + |
| *Zostera marina* | + | | | |
| *Zostera noltei* | | | ++++ | |
| Saxifragales | | | | |
| *Myriophyllum spicatum* | | | + | |
| **CHAROPHYTA** | | | | |
| **Charales** | | | | |
| *Chara baltica* | | | | +++ |
| *Chara contraria* | | | + | |
| *Lamprothamnium papulosum* | | | | + |
| *Lamprothamnium succinctum* | | | + | |
| **Zygnematales** | | | | |
| *Spirogyra sp.* | | | | ++++ |
| *Zygnema sp.* | | | | + |
| **CHLOROPHYTA** | | | | |
| **Bryopsidales** | | | | |
| *Bryopsis cupressina* | | | + | |
| *Bryopsis hypnoides* | | | + | |
| *Bryopsis secunda* | | | + | |

**Table 2.** *Cont.*

|  | HH | EH | PH | MH |
|---|---|---|---|---|
| *Caulerpa cylindracea* |  | + |  |  |
| *Caulerpa prolifera* |  | + |  |  |
| *Codium bursa* |  | + |  |  |
| *Derbesia tenuissima* |  | + | + | + |
| *Halimeda tuna* |  | + |  |  |
| *Pedobesia simplex* |  |  |  | + |
| **Cladophorales** |  |  |  |  |
| *Aegagropila linnaei* | + |  | + | + |
| *Anadyomene stellata* |  | ++ |  |  |
| *Chaetomorpha aerea* |  | ++ | + |  |
| *Chaetomorpha ligustica* |  |  | + |  |
| *Chaetomorpha linum* | + | ++ | + |  |
| *Chaetomorpha tortuosa* | + | + | + | + |
| *Cladophora albida* |  | + | + |  |
| *Cladophora coelothrix* |  |  | + |  |
| *Cladophora dalmatica* | + | + | + |  |
| *Cladophora fracta* |  | + | + | + |
| *Cladophora glomerata* |  | + | + | + |
| *Cladophora hutchinsiae* |  | + |  |  |
| *Cladophora laetevirens* |  | + |  |  |
| *Cladophora lehmanniana* |  | + | + |  |
| *Cladophora liniformis* | + |  | + |  |
| *Cladophora prolifera* |  | + | + |  |
| *Cladophora rupestris* |  | + | + |  |
| *Cladophora sericea* | ++ |  | + |  |
| *Cladophora vadorum* | + | + | + | ++ |
| *Cladophora vagabunda* | ++ | + | + |  |
| *Cladophoropsis membranacea* |  | + |  |  |
| *Lychaete echinus* |  | + | + |  |
| *Rhizoclonium riparium* |  |  | + |  |
| *Valonia macrophysa* |  | + |  |  |
| *Valonia utricularis* |  | ++ |  |  |
| **Dasycladales** |  |  |  |  |
| *Acetabularia acetabulum* |  | + |  |  |
| *Dasycladus vermicularis* |  | + |  |  |
| **Ulotrichales** |  |  |  |  |
| *Ulothrix flacca* |  |  | + |  |
| *Ulothrix implexa* | + | + | + |  |
| *Urospora penicilliformis* | ++ | + | + |  |
| **Ulvales** |  |  |  |  |
| *Blidingia marginata* | + |  |  |  |
| *Blidingia minima* |  | + |  |  |

**Table 2.** *Cont.*

|  | HH | EH | PH | MH |
|---|:---:|:---:|:---:|:---:|
| *Ulva australis* | + | + | + |  |
| *Ulva clathrata* | + |  |  | + |
| *Ulva compressa* | + | + | + |  |
| *Ulva curvata* | + |  | + |  |
| *Ulva flexuosa* | + | + | + |  |
| *Ulva intestinalis* | +++ | + | + | + |
| *Ulva kylinii* | + |  |  |  |
| *Ulva linza* | + |  |  |  |
| *Ulva prolifera* | ++ | + |  |  |
| *Ulva prolifera subsp. blidingiana* |  |  | + |  |
| *Ulva pseudorotundata* |  | + | + |  |
| *Ulva rigida* |  | + | + |  |
| *Ulvaria obscura* |  |  | + |  |
| *Ulvella lens* |  | + |  |  |
| *Ulvella viridis* |  | + | + |  |
| *Umbraulva dangeardii* |  | + |  |  |
| **OCHROPHYTA-Phaeophyceae** |  |  |  |  |
| **Dictyotales** |  |  |  |  |
| *Dictyota dichotoma* |  | ++ |  |  |
| *Dictyota dichotoma var. intricata* |  | + |  |  |
| *Dictyota implexa* |  | + |  |  |
| *Dictyota mediterranea* |  | + |  |  |
| *Padina ditristromatica* |  | + |  |  |
| *Padina pavonica* |  | + |  |  |
| **Ectocarpales** |  |  |  |  |
| *Ectocarpus fasciculatus* | + |  |  |  |
| *Ectocarpus siliculosus* | + |  |  |  |
| *Pylaiella littoralis* |  |  |  | + |
| *Scytosiphon lomentaria* |  | + | + |  |
| **Fucales** |  |  |  |  |
| *Cystoseira aurantia* |  |  | + |  |
| *Cystoseira compressa* |  |  | + |  |
| *Gongolaria barbata* |  | +++ | + |  |
| **RHODOPHYTA** |  |  |  |  |
| **Acrochaetiales** |  |  |  |  |
| *Acrochaetium secundatum* |  |  | + |  |
| **Bangiales** |  |  |  |  |
| *Neoyropia leucosticta* |  | + | + |  |
| **Bonnemaisoniales** |  |  |  |  |
| *Asparagopsis taxiformis* |  | ++ |  |  |
| **Ceramiales** |  |  |  |  |
| *Acrosorium ciliolatum* |  | + |  |  |
| *Aglaothamnion feldmanniae* |  |  | + |  |

**Table 2.** *Cont.*

| | HH | EH | PH | MH |
|---|---|---|---|---|
| *Aglaothamnion tenuissimum* | | | + | |
| *Aglaothamnion tripinnatum* | | | + | |
| *Alsidium corallinum* | | + | + | |
| *Alsidium helminthochorton* | | + | | |
| *Anotrichium furcellatum* | | + | | |
| *Antithamnion cruciatum* | | + | | |
| *Carradoriella denudata* | | + | + | |
| *Carradoriella elongata* | | + | | |
| *Ceramium ciliatum* | | | + | |
| *Ceramium cimbricum* | | + | ++ | |
| *Ceramium cimbricum f. flaccidum* | | + | | |
| *Ceramium codii* | | | + | |
| *Ceramium deslongchampsii* | | + | + | |
| *Ceramium diaphanum* | | + | ++ | |
| *Ceramium echionotum* | | + | + | |
| *Ceramium siliquosum* | | + | + | |
| *Ceramium tenerrimum* | | + | + | |
| *Chondria capillaris* | + | + | +++ | |
| *Chondria coerulescens* | | + | | |
| *Chondria dasyphylla* | | + | + | |
| *Chondria polyrhiza* | | + | | |
| *Dasya pedicellata* | | + | +++ | |
| *Dasya ocellata* | | + | | |
| *Gayliella mazoyerae* | | + | + | |
| *Griffithsia opuntioides* | | + | | |
| *Griffithsia schousboei* | | + | | |
| *Herposiphonia secunda* | | + | + | |
| *Heterosiphonia crispella* | | + | | |
| *Hypoglossum hypoglossoides* | | + | | |
| *Laurencia obtusa* | | + | | |
| *Leptosiphonia fibrillosa* | + | + | + | |
| *Lophosiphonia cristata* | | + | | |
| *Lophosiphonia obscura* | + | + | + | + |
| *Melanothamnus harveyi* | | + | + | |
| *Osmundea oederi* | | + | | |
| *Palisada perforata* | | + | | |
| *Palisada thuyoides* | | + | | |
| *Polysiphonia atlantica* | | + | + | |
| *Polysiphonia opaca* | | + | | |

**Table 2.** *Cont.*

| | HH | EH | PH | MH |
|---|---|---|---|---|
| *Polysiphonia stricta* | ++ | | + | |
| *Pterothamnion crispum* | | + | | |
| *Rytiphlaea tinctoria* | | + | | |
| *Spyridia filamentosa* | | ++ | + | |
| *Vertebrata fruticulosa* | | + | | |
| *Vertebrata fucoides* | | + | | |
| *Vertebrata furcellata* | + | + | + | |
| *Vertebrata reptabunda* | | + | | |
| *Wrangelia penicillata* | | + | | |
| **Colaconematales** | | | | |
| *Colaconema caespitosum* | | | + | |
| *Colaconema corymbiferum* | | | | + |
| *Colaconema daviesii* | | + | + | |
| **Corallinales** | | | | |
| *Amphiroa rigida* | | + | | |
| *Corallina officinalis* | | + | | |
| *Ellisolandia elongata* | | + | | |
| *Hydrolithon cruciatum* | | + | + | |
| *Hydrolithon farinosum* | | + | + | |
| *Jania rubens* | | + | | |
| *Jania virgata* | | + | | |
| *Melobesia membranacea* | | + | + | |
| *Pneophyllum fragile* | | + | | |
| *Titanoderma pustulatum* | | | + | |
| **Erythropeltales** | | | | |
| *Erythrotrichia carnea* | | + | + | |
| **Gelidiales** | | | | |
| *Gelidium crinale* | | + | + | |
| *Pterocladiella capillacea* | | + | | |
| **Gigartinales** | | | | |
| *Caulacanthus ustulatus* | | + | | |
| *Chondracanthus acicularis* | | + | | |
| *Chondracanthus teedei* | | + | | |
| *Feldmannophycus rayssiae* | | + | | |
| *Gymnogongrus griffithsiae* | | + | + | |
| *Hypnea corona* | | + | | |
| *Hypnea musciformis* | | + | | |
| *Hypnea spinella* | | + | | |
| *Schmitziella endophloea* | | + | | |
| *Wurdemannia miniata* | | + | + | |
| **Gracilariales** | | | | |
| *Gracilaria bursa-pastoris* | | + | | |
| *Gracilaria gracilis* | | ++ | ++ | |

**Table 2.** *Cont.*

|  | HH | EH | PH | MH |
|---|---|---|---|---|
| *Gracilaria longa* |  |  | + |  |
| **Halymeniales** |  |  |  |  |
| *Grateloupia filicina* |  | + | + |  |
| **Rhodymeniales** |  |  |  |  |
| *Champia parvula* |  | + | + |  |
| *Chylocladia verticillata* |  |  | + |  |
| *Lomentaria articulata* |  |  | + |  |
| *Rhodymenia ardissonei* |  | + |  |  |
| *Rhodymenia sp.* |  | + |  |  |
| **Stylonematales** |  |  |  |  |
| *Stylonema alsidii* |  |  | + |  |

The Species Richness (S) of macrophyte assemblages significantly varied (*p*-value: 0.001) according to the salinity class of T-WBs (Figure 2a,b). Salinity values at the end of the range generally reflected a scarce macrophyte diversity, as in the case of the mesohaline T-WBs of TG and CE and the hyperhaline PU. LS, similarly to hyperhaline, showed a moderate macrophyte diversity mainly represented by opportunistic species. On the other hand, macrophyte assemblages proved to be much more diverse in polyhaline (LE1, LE2, LE3 and VA) and euhaline T-WBs (MP1, MP2 and PC). The strongest reduction in Species Richness was observed in Rhodophyta, for which the number of species consistently dropped along the salinity gradient (75 in euhaline T-WBs vs. 5 and 2 in hyperhaline and mesohaline T-WBs, respectively), with a reduction between 93 and 97%. The percentage of the decrease in species number was still quite high in the Ochrophyta-Phaeophyceae (75–88%) and the lowest was displayed by Chlorophyta (49–77%).

The Cluster Analysis demonstrated that T-WBs within the same salinity class had remarkable analogies in macrophyte species composition and frequency, forming four well-distinct groups with a high level of similarity, especially in the case of hyperhaline, mesohaline and polyhaline T-WBs (Figure 3a). On a smaller scale, the Cluster Analysis showed the existence of two groups of T-WBs. In fact, the clustering highlighted a certain similarity (~17%) between macrophyte assemblages occurring at both the ends of the salinity range (mesohaline and hyperhaline T-WBs), as well as between those of polyhaline and euhaline T-WBs (similarity ~16%). The *n*–MDS confirmed the presence of four distinct groups corresponding to the four salinity classes, with a stress value of 0.14 (Figure 3b). The role of the salinity regime in influencing macrophyte composition was confirmed by ANOSIM, which gave back a Global *R*-value of ~0.95 and a *p*-value of 0.002, attesting to highly significant differences among assemblages (Table 3). However, although the pairwise ANOSIM revealed very high *R*-values, only the comparison between euhaline and polyhaline T-WBs was demonstrated to be strictly significant (*p*-value < 0.05) because of the scarce number of species counted in hyperhaline and mesohaline T-WBs, as well as due to the presence of several eurytherm and euryhaline species found in most of the analyzed T-WBs.

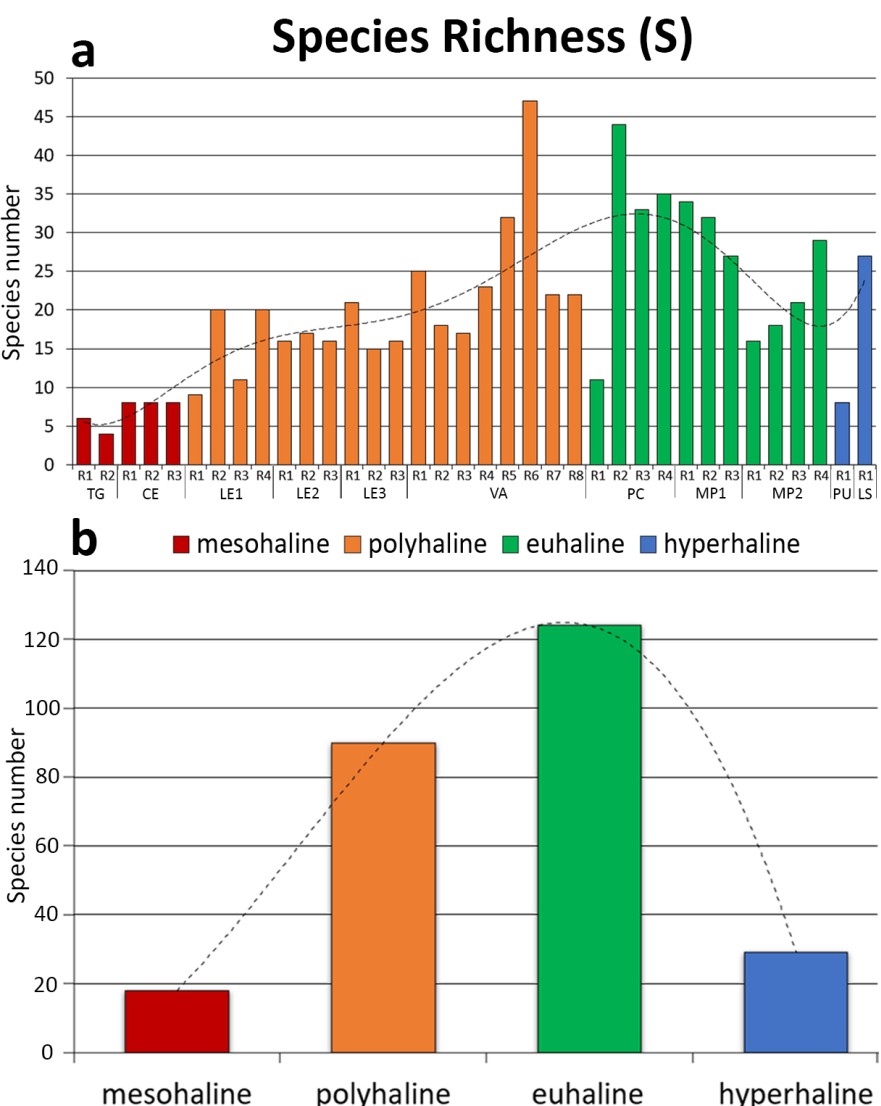

**Figure 2.** Trend of the species richness (i.e., number of species) of each sampling station (R) in relation to the salinity level of each T-WB considered in the study (**a**) and total number of species counted for each of the four classes (**b**).

**Table 3.** Pairwise and global results of the ANOSIM test carried out to verify the data significance.

| Pairwise ANOSIM | *R*-Values | *p*-Values |
|---|---|---|
| hyperhaline vs. euhaline | 1 | 0.1019 |
| hyperhaline vs. polyhaline | 1 | 0.0649 |
| hyperhaline vs. mesohaline | 1 | 0.3290 |
| euhaline vs. polyhaline | 0.9630 | 0.0278 * |
| euhaline vs. mesohaline | 1 | 0.0986 |
| polyhaline vs. mesohaline | 1 | 0.0659 |
| **Global ANOSIM** | *R*-values | *p*-values |
|  | 0.9545 | 0.0002 *** |

* $p < 0.05$; *** $p < 0.001$.

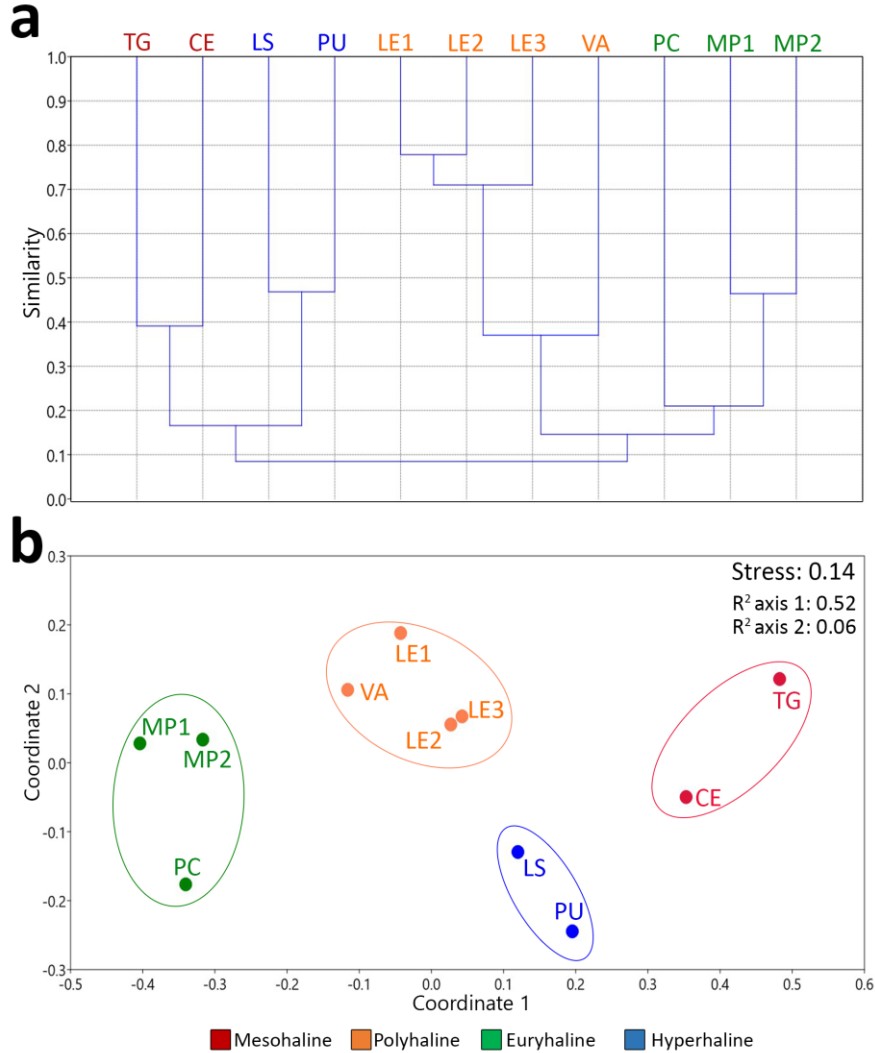

**Figure 3.** Results of Cluster Analyses (**a**) and *non*−Metric Multidimensional Scaling (*n*−MDS) (**b**) based on Bray–Curtis dissimilarities between T-WBs of the four salinity classes.

The polyhaline T-WBs of LE1, LE2, LE3 and VA showed the highest Group Average Similarity (55.15%), with high resemblances in terms of species composition and frequency (Table 4). The hyperhaline T-WBs of LS and PU showed a high Group Average Similarity of 46.85%, given by the restricted number of species. The mesohaline T-WBs of CE and TG shared few taxa with a Group Average Similarity of 39.11%, while the euhaline T-WBs, including PC, MP1 and MP2, showed a great number of common species but with a Group Average Similarity of 29.49%. The macrophyte composition and species mean abundance of the mesohaline T-WB-type was dominated by Charophyta. In fact, almost 70% of the similarity was due to the contribution of a species of *Spirogyra* (only identified at a generic level) and *C. baltica*, both characteristic of lower salinity waters. About 32% of this similarity was contributed by *Cladophora vadorum* (Areschoug) Kützing, a species with a wide distribution in lagoonal environments. Macrophyte composition of the polyhaline T-WB-type was dominated by the angiosperm *Z. noltei* and by different species of Ceramiales such as *Chondria capillaris* (Hudson) M.J. Wynne, *Dasya pedicellata* (C. Agardh) C. Agardh and various *Ceramium* spp. which, together, contributed up to about 80% of the total similarity within this type. In the hyperhaline T-WB-type, the macrophyte composition was dominated by *R. spiralis* and two species of Ulvales, *Ulva intestinalis* Linnaeus and *U. prolifera* O.F. Müller, reaching a total of 76.5% of the similarity within this type. Macrophyte composition of the euhaline T-WB-type was composed by an array of heterogeneous taxa,

which was not only typical of marine waters, such as some brown algae belonging to Fucales and Dictyotales, but also more ubiquitous species that mainly belong to Ceramiales and Cladophorales (Table 4).

**Table 4.** SIMPER results showing the average abundance, average similarity, percentage contribution of each species in similarity among the four salinity classes, percentage cumulative similarity and average similarity of the groups.

| | TAXON | Av. Abund. | Av. Sim. | Contrib. % | Cum. % | Group Av. Sim. % |
|---|---|---|---|---|---|---|
| **MESO HALINE** | *Spirogyra sp.* | 80.92 | 22.76 | 58.20 | 58.20 | |
| | *Cladophora vadorum* | 40.83 | 12.41 | 31.75 | 89.95 | 39.11 |
| | *Chara baltica* | 56.33 | 3.93 | 10.05 | 100.00 | |
| **POLYHALINE** | *Zostera noltei* | 76.66 | 12.49 | 22.65 | 22.65 | |
| | *Chondria capillaris* | 63.03 | 10.92 | 19.79 | 42.45 | |
| | *Dasya pedicellata* | 56.25 | 8.60 | 15.60 | 58.05 | |
| | *Ceramium cimbricum* | 32.04 | 4.31 | 7.82 | 65.86 | |
| | *Ceramium diaphanum* | 30.61 | 3.82 | 6.93 | 72.79 | 55.15 |
| | *Ceramium deslongchampsii* | 23.91 | 3.73 | 6.77 | 79.57 | |
| | *Ruppia spiralis* | 37.50 | 2.50 | 4.53 | 84.10 | |
| | *Gracilaria gracilis* | 26.35 | 1.63 | 2.96 | 87.06 | |
| | *Cladophora vadorum* | 7.81 | 1.40 | 2.53 | 89.59 | |
| **EUHALINE** | *Gongolaria barbata* | 50.06 | 3.23 | 10.97 | 10.97 | |
| | *Cymodocea nodosa* | 51.19 | 3.18 | 10.79 | 21.76 | |
| | *Valonia utricularis* | 37.50 | 3.00 | 10.16 | 31.93 | |
| | *Chaetomorpha aerea* | 44.58 | 2.74 | 9.29 | 41.22 | |
| | *Ceramium deslongchampsii* | 23.75 | 2.67 | 9.06 | 50.28 | |
| | *Dictyota dichotoma* | 31.81 | 2.34 | 7.93 | 58.21 | |
| | *Spyridia filamentosa* | 32.08 | 2.16 | 7.31 | 65.52 | |
| | *Gracilaria gracilis* | 26.94 | 1.55 | 5.27 | 70.80 | 29.49 |
| | *Cladophora prolifera* | 19.03 | 1.11 | 3.76 | 74.55 | |
| | *Chaetomorpha linum* | 17.94 | 0.92 | 3.13 | 77.69 | |
| | *Hypnea corona* | 13.47 | 0.78 | 2.64 | 80.32 | |
| | *Rhodymenia ardissonei* | 16.03 | 0.64 | 2.17 | 82.50 | |
| | *Cladophora vagabunda* | 4.58 | 0.52 | 1.77 | 84.26 | |
| | *Cladophora laetevirens* | 6.11 | 0.49 | 1.65 | 85.92 | |
| | *Ceramium diaphanum* | 4.72 | 0.41 | 1.40 | 87.32 | |
| | *Gayliella mazoyerae* | 3.89 | 0.34 | 1.17 | 88.48 | |
| | *Carradoriella denudata* | 2.78 | 0.32 | 1.07 | 89.55 | |
| **HYPERHALINE** | *Ruppia spiralis* | 100.00 | 20.02 | 42.74 | 42.74 | |
| | *Ulva intestinalis* | 57.50 | 10.01 | 21.37 | 64.10 | |
| | *Ulva prolifera* | 42.00 | 5.81 | 12.39 | 76.50 | 46.85 |
| | *Urospora penicilliformis* | 28.00 | 5.61 | 11.97 | 88.46 | |
| | *Polysiphonia stricta* | 27.00 | 5.41 | 11.54 | 100.00 | |

## 4. Discussion

The definition of surface T-WB types is the first milestone in the implementation of the WFD (Section 1.2, Annex II) [21] to enable the establishment of appropriate reference conditions and to make a valid assessment of the ecological status of T-WBs. However, at the European level, many difficulties have been encountered in defining a general typology for transitional waters [46]. In particular, in the Mediterranean ecoregion, typologies based on different descriptors have been proposed, e.g., [40,45–50], but a common consensus is still lacking. An approach based on the composition of macrophyte assemblages was used in this study to test the validity of the Apulian transitional water types, considering that they are representative of other Mediterranean ecosystems. Our results highlighted that a typology based on salinity ranges accounts for the variability of macrophyte assemblages, as required by WFD. Salinity proved to be the main environmental factor driving both the distribution and the composition of macrophyte communities, confirming previous observations in other areas [7,10,51–58]. In fact, although several geomorphological and hydrological factors can also play a determinant role in modeling macrophyte assemblages [59,60], a salinity regime well-explained the variability found in the T-WBs along the gradient of salinity. This agrees with Pèrez-Ruzafa et al. [1], who had also identified the salinity regime as an influential factor, highly affecting sessile organisms such as macrophytes and being one of the main drivers in modeling macrophyte biodiversity in transitional water bodies. Despite the differences in morphology, depth and extension, as well as spatial distances among T-WBs, those within the same salinity class demonstrated comparable macrophyte communities in most of the cases. The mesohaline community was mainly represented by the Charophyta *Spirogyra* sp. and *C. baltica*, which demonstrated high selectivity for the mesohaline T-WBs (TG and CE), as they are only found in both the basins although they represent quite different environments in terms of their physico-chemical features. In particular, the genus *Spirogyra* confirmed its preference for nearly freshwater habitats [61], while *C. baltica* demonstrated the ability to tolerate brackish waters with salinity up to 15 PSU [62].

On the other hand, geomorphologically and hydrologically similar coastal lagoons, but with different salinity classes, demonstrated significant differences in their macrophyte composition, and, once again, salinity played a crucial role in influencing macrophyte assemblages, as well as in the classification of lagoon types. The polyhaline community was represented by the dominance of *Z. noltei*, a species considered euryhaline [63], which formed monospecific beds or coexisted with *R. spiralis*. This angiosperm also characterized macrophyte assemblages of meso- to polyhaline waters of the Rodia lagoon in Western Greece [27]. The macrophyte community of the euhaline type was mainly represented by the euryhaline *C. nodosa* [54] and other typical marine species, such as the brown algae *G. barbata* and *Dictyota dichotoma* (Hudson) J.V. Lamouroux, the green *Valonia utricularis* (Roth) C. Agardh and the red *Spyridia filamentosa* (Wulfen) Harvey, which are well adapted to a salinity range that is almost the same as the open sea. Among the most abundant species, *G. barbata* and *C. nodosa* proved to be widely spread in the euhaline T-WBs (PC, MP1 and MP2), forming secondary substrates for several species and enhancing biodiversity. Even though they were harboring some common marine taxa, the euhaline coastal lagoons examined in this study displayed the lowest average similarity and some differences in macrophyte spectra composition that can be related to morphological aspects, such as the shape of the lagoons, the depth of the basins and the sea/lagoon exchange features. For instance, PC is an almost elliptical bay (2 km$^2$), only partially delimited by a narrow peninsula which leaves a permanent connection with the open sea through a channel system about 700 m wide; its average depth is about 1 m [64]. MP is over ten times larger than PC, with a surface area of 20.72 km$^2$, but it is a semi-enclosed basin divided into the First Inlet (MP1 T-WB) and Second Inlet (MP2 T-WB), with a maximum depth of 12 and 8 m, respectively. However, only the First Inlet is in communication with the Ionian Sea through two channels, thus causing a reduced water exchange [65]. Effectively, PC was almost exclusively colonized by marine taxa, most of which shared with the adjacent open sea assemblages [66], indicating

constant species recruitment from these habitats. On the contrary, MP1 and, especially, MP2, were also inhabited by species of more confined environments, such as *Chaetomorpha aerea* (Dillwyn) Kützing and *Ulva rigida* C. Agardh. The macrophyte community characterizing the hyperhaline type was mainly represented by *R. spiralis*. The genus *Ruppia* has long suffered a chaotic taxonomy, but recently the three species *R. maritima* Linnaeus, *R. cirrhosa* (Petagna) Grande and *R. spiralis* Linnaeus *ex* Dumortier have been considered as fully independent taxa [67]. These authors underlined that many Mediterranean records of *R. cirrhosa* might actually be referring to *R. spiralis*, a more common European species occurring in brackish habitats. Our specimens matched the specific characteristics of this species; therefore, they are here reported as *R. spiralis*. This species can be considered as the most "marine" *Ruppia* in Europe [67], with a broader salt tolerance. The presence of very dense and extensive meadows of *R. spiralis* in hyperhaline T-WBs, as well as in polyhaline T-WBs of the Lesina lagoon, could be related to the high occurrence of sexual reproduction, which is known to strongly enhance the spread of these plants over large distances. This is particularly true for PU, where this phanerogam was remarkably abundant all over the T-WB. The frequent occurrence of sexual reproduction was testified by the high number of flowers observed during our analysis. The other aquatic phanerogams found during our study were demonstrated to mostly rely on asexual reproduction for the colonization of T-WBs, as testified by the total absence of flowers and/or fruits for *C. nodosa* [68] and by their occasional presence in the case of *Z. noltei* [69].

In addition to the main species characterizing the different macrophyte communities, some opportunistic taxa were revealed to be well adapted to salinity variations and occurred in all the salinity ranges. They mainly belonged to the orders Ulvales and Cladophorales (Chlorophyta) and Ceramiales (Rhodophyta).

In transitional waters, other factors, such as trophic status and water oxygenation, may drive the macrophyte community structure [7]. However, the TWEAM multi-index method for the eutrophication assessment in these ecosystems [70] classified over 60% of the water bodies examined in this study in the oligothropic status, except for LE1, MP1, MP2 and TG, which were mesotrophic.

The Species Richness index was closely associated with salinity. Macrophyte diversity was the highest at sampling stations of euhaline T-WBs, which had salinity levels similar to marine waters, started to decrease at sampling stations of polyhaline T-WBs and reached minimum values in correspondence with sampling stations placed at the two ends of the salinity range. This declining trend in the number of taxa, depending on too low or too high salinity, was already reported in Mediterranean coastal lagoons [19,54,71] and in the Baltic Sea [72], and it was confirmed in the present study. The lowest species richness recorded in mesohaline and hyperhaline T-WBs, where salinity was quite far from marine waters, reflected stress conditions, with fewer species tolerating these extremes. This was also observed by Sfriso et al., 2017 [73], who demonstrated that salinity degrees similar to those of the open sea enhanced macrophyte biodiversity, while extreme values determined a reduction in Species Richness. Besides the crucial role of salinity, the topology of T-WBs, also involving their degree of fragmentation and their size, may influence the Species Richness of macrophytes. In fact, we observed that the sampling stations placed in uninterrupted and/or wider T-WBs (such as LE1, LE2, LE3, VA, PC, MP1 and MP2) had a higher number of macrophyte species with respect to sampling stations placed in patchy and smaller T-WBs (this is the case of TG, PU and CE), confirming the role of habitat fragmentation and habitat size in enhancing biodiversity, as proved in previous studies [74,75]. Moreover, the degree of confinement of T-WBs, besides influencing the qualitative composition of macrophyte assemblages (see above), can also play a role in affecting Species Richness, even at a sampling station scale. For instance, sampling stations placed at MP1, which directly communicate with the open sea, demonstrated higher levels of Species Richness than those of MP2, which is a more confined T-WB (Figure 1). MP1 and MP2 were also the deepest among the T-WBs examined in this study. Considering that both light quality and quantity depend on depth, this factor could also contribute to

structure macrophyte distribution and influence Species Richness, especially with regards to red algae that are more adapted to lower irradiance and usually thrive best in deeper waters [7]. For most of them, light defines the lower limit of their depth distribution, with rare exceptions [76]. The sampling stations in MP1 and MP2 actually counted the highest number of Rhodophyta.

Rhodophyta also demonstrated to be the most "marine" group, since very few species were observed in T-WBs with salinity ranges different from the euhaline T-WBs. The Ochrophyta-Phaeophyceae also demonstrated a high preference for euhaline T-WBs, while Chlorophyta were the most salinity-adapted algal group. This trend confirmed what was observed in previous studies. For instance, Adams et al. (1999) [77] found that in South African estuaries red and brown macroalgae generally prevailed in seawater salinity conditions, whereas green algae (especially filamentous forms) were much more widespread along the whole length of estuaries. In our case, this pattern was also evident at the order level. For example, some typical marine taxa, belonging to Corallinales or Gelidiales orders, not only were found in euhaline T-WBs but were also able to tolerate the lower salinity of polyhaline T-WBs; they completely disappeared in mesohaline and hyperhaline T-WBs that were almost exclusively inhabited by opportunistic Ceramiales. The salinity ranges in these environments prevented a large colonization by red algae and favored the settlement of Chlorophyta, mainly Cladophorales and Ulvales, which were the best adapted to low-salinity and hyper-saline conditions [78,79]. Among Ochrophyta-Phaeophyceae, likewise, well-structured Dictyotales and Fucales species were found in euhaline and polyhaline T-WBs, while very few opportunistic species of Ectocarpales were the only ones present in the other T-WBs.

## 5. Conclusions

Macrophyte assemblages of T-WBs were demonstrated to be influenced by salinity both in terms of species composition and richness, testifying to the crucial role of this abiotic factor in modeling such communities. However, even though the macrophyte variability found in T-WBs was well-explained by the salinity regime, other abiotic factors may contribute to structuring these assemblages and further studies should be addressed to understand their importance in influencing the macrophyte presence and distribution.

This study also contributed to implementing the knowledge about the macrophyte assemblages of the most important T-WBs of the Apulian region, whose biodiversity was still scarcely known in most of the cases. The assessment of macrophyte diversity and the structure of their assemblages in transitional waters is a key point for the development and implementation of management strategies, which represent one of the main goals of WFD. At the same time, the periodic monitoring of the conservation status of these ecosystems, based on biological indicators such as macrophytes, allows one to understand the effectiveness of such management actions, with the aim to reach and/or maintain the Good Ecological Status claimed by WFD.

**Author Contributions:** Conceptualization, A.T., F.M. and A.B.; methodology, A.T., A.L., F.M., N.U. and A.B.; software, A.T., F.M. and A.B.; validation A.T., F.M., G.C. and A.B.; formal analysis, A.T., A.L., F.M., N.U. and A.B.; investigation, A.L., N.U. and A.B.; resources, A.B.; data curation, A.T., F.M. and A.B.; original draft preparation, A.T. and A.B.; writing—review and editing, A.T., A.L., F.M., G.C., N.U. and A.B.; visualization, A.T. and A.B.; supervision, A.B.; project administration, A.B.; funding acquisition, A.B. All authors have read and agreed to the published version of the manuscript.

**Funding:** This research was funded by the Apulian Regional Agency for the Environmental Prevention and Protection, Research Agreement "Monitoring of Biological Quality Elements (BQEs) macroalgae and phanerogams in the Apulian transitional waters. CUP: B31G22000110002.

**Institutional Review Board Statement:** Not applicable.

**Data Availability Statement:** The dataset generated during and/or analyzed for the current study is available from the corresponding author upon request.

**Conflicts of Interest:** The authors declare no conflict of interest. The funders had no role in the design of the study; in the collection, analyses, or interpretation of data; in the writing of the manuscript; or in the decision to publish the results.

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
