# Peer review of "Salinity as a Key Factor in Structuring Macrophyte Assemblages in Transitional Water Bodies: The Case of the Apulian Coastal Lagoons (Southern Italy)"

_diversity, doi:10.3390/d15050615_

Round 1
Reviewer 1 Report
Review of the manuscript: Diversity 2312671
Salinity as a key factor in structuring macrophyte assemblages 2 in transitional water bodies: the case of the Apulian coastal la-3 goons (Southern Italy)
General comments
This study focuses on the effect of salinity on submerged macrophyte assemblages in 11 transitional water bodies from 8 coastal lagoons of Apulia region (Italia). All these water bodies are located in non-tidal coastal lagoons of small and medium size, classified as mesohaline, polyhaline, euhaline and hyperhaline. The authors aim at proving the role of salinity as a structuring factor for macrophyte communities. They assess a significant relationship between richness index and salinity class and also identify 4 distinct macrophyte communities strongly related to salinity class.
The paper is very well written. The material and methods used are adapted to the biological questions and correctly described. The results are properly presented and the discussion is well documented.
The link between species assemblages and salinity classes has been correctly studied using cluster analysis, n-MDS, ANOSIM and SIMPER test. However, concerning the relationship between species richness and salinity, there is a bias due to a higher sampling effort in polyhaline and euhaline T-WB (from 3 to 8 stations) vs. mesohaline and polyhaline ones (1 to 3 stations). It is well known that species richness increases with number of samples and in the present case, the higher richness observed in poly- and euhaline T-WB can also be imputed to this higher sampling effort. Consequently, I recommend that Fig 2 shows one bar per station (and not per T-WB or salinity class) in order to compare items with the same sampling effort. In addition the significance of the relationship between the number of species in a given station and the salinity class should be assessed using a statistical test, not only by describing the smoothed trend on the graph. The discussion paragraph about species richness (l. 298...) and the conclusion have to be checked according to the new results.
Could the authors consider a slightly more structured discussion (e.g. smaller paragraphs) to enhance the readability? This is only a suggestion...
Specific comments
l. 136: Please specify which aggregating method was applied to the cluster analysis (Ward ? Average link ?...)
l. 191: The p-value is not correct in : “only the comparison between euhaline and 190 polyhaline T-WBs showed to be strictly significant (p-value < 0.005)” => p-value < 0.05
l. 191 to 194: “probably because of…” : this is interpretation of the results and should be moved to the discussion section.
l. 307… : numbers and percentages of species should have been given in the result section.
Reviewer 2 Report
Coastal lagoon is one important ecosystem of transitional waters, characterized by high fluctuations in water phisicochemical and biological conditions. This study is focused on the effect of salinity on macrophyte communities (as an important primary producer) of the eight coastal lagoons of Apulia (south-eastern Mediterranean Sea). Detailed information on the surveyed sites were given on macrophyte occurrence, biodiversity and its relationship with environmental parameters. The authors did a very nice work on collecting the limnological information and conducted a sound analysis revealing the macrophyte communities changes along a significant salinity gradient.
I just have one question. As we know there are lots of regulators on aquatic plants, including water depth, hydrological fluctuations, nutrient loading, salinity and so on. I agree with the macrophyte difference over the salinity gradient but I am curious about effect from other environmental variables. I suppose the author also conducted relevant survey on environmental variables being listed above. Therefore I strongly suggest to add one quantitative correlation coefficient table showing this possible colinear impacts from multiple-stressors. On such basis, the authors can have a more comprehensive discussion on the possible cofounding effects from other variables, besides the salinity driver.
Reviewer 3 Report
· please state paradigms of macrophyte species abundance variation on fitness traits useful for species on their edge of distribution
· clarify the statistic consistency of homogeneous ecological conditions (if any) when predicting sample heterogeneity with high prediction occurrence
· do you suggest possible linkage of merging data & resources between data banks of similar macrophyte null hypotheses?
· discuss if angiosperms / phanerogams reproductive strategy (sexual vs asexual) with ramets & gamets play a key role?
· fig 2 is missing ± SD info; better presented in box-plots with Tukey test?
· discuss validity of best fitted cluster analysis scaffold to your dataset
· state % fitness of coordinates in your multivariate analysis
· discuss Med topology ecoregions with their peculiarity on SR
· fragmented habitat patchiness in transitional waters influence SR?
· bibliography to be utilized on the light of the above discussion points:
1. Robinson BJO, Morley SA, Rizouli A, Sarantopoulou J, Gkafas GA, Exadactylos A, Küpper FC (2022) New confirmed depth-limit of Antarctic macroalgae: Palmaria decipiens found at 100m depth in the Southern Ocean. Polar Biology 45 (8): 1459-1463
2. Konstantinidis I, Gkafas GA, Papathanasiou V, Orfanidis S, Küpper FC, Arnaud-Haond S, Exadactylos A (2022) Biogeography pattern of the marine angiosperm Cymodocea nodosa in the eastern Mediterranean Sea related to the quaternary climatic changes. Ecology and Evolution 12 (5): e8911
Reviewer 4 Report
The authors set up a long term research and did get some very good data. A sound relationship between species richness and salinity was described in the paper. And I think it will be even better if the authors can support some additional data to exclude the influence from other environmental issues, like water depth, nutrients etc., though it is not obligatory.
For the writing, I am not sure if it’s appropriate to use the word “macrophyte” to enclose all the plants and algae the authors found, especially when the later represented a major diversity, which make me confused. I hope the authors may find some more accurate word instead of macrophytes.
The references are generally good, with only a few (e.g. 35#) looks not in English, and may be difficult for international readers.
Round 2
Reviewer 4 Report
The authors have clearly replied my questions and improved the writing as I suggested. I think the paper is pretty good now and can be published.